# The Latest Craniofacial Reconstructive Techniques Using Anchored Implants after Surgical Treatment of Nasal and Paranasal Sinuses Tumors

**DOI:** 10.3390/healthcare11121663

**Published:** 2023-06-06

**Authors:** Karolina Dżaman, Marlena Ziemska-Gorczyca, Igor Anurin, Magdalena Błaszczyk

**Affiliations:** 1Department of Otolaryngology, Centre of Postgraduate Medical Education, Marymoncka 99/103, 01-813 Warsaw, Poland; 2Faculty of Science and Technology, University of Silesia in Katowice, 40-007 Katowice, Poland

**Keywords:** nasal prosthesis, osseointegrated implants, craniofacial reconstruction, sinonasal tumor, radiotherapy, extraoral implantology

## Abstract

Reconstructive surgery after surgical treatment of neoplasms in the head and neck region is always a challenge. Many factors are responsible for the success of reconstruction. The anatomy of the facial region is complex, which significantly influences the aesthetic effect of the reconstruction. Moreover, many patients undergo postoperative radiotherapy after surgical treatment, which affects the range of reconstructive techniques. The aim of this study is to review current reconstructive methods in the craniofacial region, using bone-anchored implants to attach nasal prostheses. The article also comprises the authors’ own experience with successful single-stage, Vistafix 3 osseointegrated implants for the attachment of an external nasal prosthesis in a 51-year-old man after surgical removal of squamous cell carcinoma of the nose and paranasal sinuses. The literature search for articles regarding implants in craniofacial reconstructions was performed using the three following databases: Scopus, Web of Science and MEDLINE (through PubMed), and follows the Preferred Reporting Items for Systematic Reviews and Meta-Analyses Statement (PRISMA). A systematic literature search was set for 2018–2023 and retrieved 92 studies. From them, 18 articles were included in the review.

## 1. Introduction

Malignant neoplasms of paranasal sinuses account for 3–5% of tumors detected in the head and neck region [1,2]. Squamous cell carcinoma (SCC) of the nasal cavity and sinuses is one of the more common malignant tumors of this area. SCC constitutes 50–60% of sinonasal cancers [3]. It is an aggressive and insidious cancer that is asymptomatic at a low stage. Due to its late and non-specific symptoms, most patients present to the doctor at a late stage of the disease. Even 72.8% of patients with sinonasal cancer come to the doctor in the advanced stage of the disease (stage 3 and 4 UICC) [4,5]. The treatment of choice is surgery, and if the surgery is not radical, patients are referred for adjuvant radiotherapy. In patients at a very advanced stage, neoadjuvant chemotherapy offers a good chance of organ preservation and improves the outcome [6].

Radical surgical treatment requires extensive tumor removal leading to permanent deformities in the midface. The operation leads to dysfunction of the nose and sinuses, but also to large aesthetic defects, preventing the patient from returning to social life. Reconstructive surgery in the facial region poses many difficulties. Pedicle flaps displaced from the surrounding regions of the face do not have a favourable effect due to their unsatisfactory cosmetic effect but also due to additional facial scarring at the site where the flap was taken. Transplantation using the patient’s own tissues often proves impossible because the structure of the nose is made up of histologically diverse tissues, i.e., bone, cartilage, skin and the subcutaneous layer or muscle tissue, making it impossible to fully recreate the anatomy of this region. Therefore, increasingly perfect methods are being sought to design prosthetic craniofacial elements. Until recently, patients could only be offered prostheses mounted on spectacles, the use of which was not comfortable for the patient.

The aim of this article is to review current reconstructive methods in the craniofacial region, using bone-anchored implants to attach nasal prostheses. The article also describes the authors’ own experience with the first Polish successful application of single-stage, third-generation Vistafix osseointegrated implants for the attachment of an external nasal prosthesis in a 51-year-old man after surgical removal of squamous cell carcinoma of the nose and paranasal sinuses.

## 2. Methods and Materials

This systematic review follows the Preferred Reporting Items for Systematic Reviews and Meta-Analyses Statement (PRISMA).

The literature search for articles regarding implants in craniofacial reconstructions was performed using the three following databases: Scopus, Web of Science and MEDLINE (through PubMed). The last search was conducted on 15 May 2023, in each of the databases. It was not necessary to contact the authors of the retrieved research articles for additional information. Time descriptors had been set for 2018–2023. Only original research articles, case studies and reviews concerning nasal reconstruction and published in English were included. Book chapters, conference papers and opinions were excluded. Duplicates were removed using the automatic EndNote 20 duplicate finder, followed by a manual search.

## 3. Results

A systematic literature search retrieved 92 studies. From 92 records, 18 articles were included in the review. Details regarding the selection process are summarized in a custom-built PRISMA flow chart in Figure 1. Basic data on the research works included in this systematic review are collected in Table 1. Moreover, this paper also presents a clinical case of a 51-year-old male patient after rhinectomy with a nasal prosthesis mounted on anchored implants.

### 3.1. Prostheses Mounted on Anchored Implants

Due to the poor cosmetic and functional effect of the spectacle-mounted nasal prostheses, more stable methods of prosthetic attachment are being sought. For several years, titanium implants screwed into the temporal bones have been used to reconstruct auricular deformities. Similar solutions have also been attempted in other regions of the face, such as orbit [23].

Originally, the basis of craniofacial prostheses was the original Brånemark implant, which has been modified over the years. Nowadays, since 2010, self-tapping implants in unchanged form have supported Vistafix System craniofacial implants (Cochlear, Goteborg, Sweden) as well as Bone Anchored Hearing Aid (BAHA) and Active Osseointegrated Steady-State Implant System (OSI) hearing aids manufactured by Cochlear.

The Vistafix System is a craniofacial prosthesis, built from a few self-tapping titanium implants that are anchored to the bone during a surgical procedure. A durable and stable solution achieved in such way is used by a prosthetist for the creation of a trabecular scaffold—substratum for acrylic and silicone prosthesis. Thanks to the available techniques and the advanced abilities of prosthetics specialists, prostheses are created in a unique manner, with imprinting of a patient’s personal characteristics (complexion, freckles, minor telangiectasia, etc.). There are isolated reports in the literature on the effects of using anchored implants in prosthetics of the external nose. It has to be emphasized that in the midfacial region, the effective anchoring of an implant is more challenging for a surgeon than in a temporal bone. It is primarily caused by a significantly thinner bone in the middle part of the craniofacial region as well as the complicated shapes of bones in this anatomical region. Additionally, the vicinity of important anatomical structures, i.e., eyeballs, the brain, cranial nerves and large vessels, is relevant.

### 3.2. Indications

The system of craniofacial implants allows for prostheses to be designed for defects in three primary regions: ear, eye and nose. The main recipients of the solution shall be the patients who require rehabilitation after oncological treatment, injuries, burns or congenital defects. An indicator for contemplation of such prosthetics is extensive facial deformities, particularly when the possibility of reconstructive plastic surgery is limited or there are contraindications for general anaesthetics, as the implantation can be performed under the local one.

This method offers high aesthetic effects depending mainly on the quality of the prosthesis and it supports a comfortable rehabilitation process [24]. A prosthesis mounted on craniofacial implants allows it to be fixed in place without the risk of displacement, and it can be put on and removed efficiently and easily.

Adhesion of the prosthesis to the face is determined by the number of anchored implants and their stability in the patient’s bone. The prosthesis can also provide support for corrective spectacles or sunglasses. It is also worth noting that facial prostheses fixed on anchored implants are tolerated better than adhesively fixed implants and, therefore, provide a greater improvement in patients’ quality of life [25,26].

It is assumed that the optimum number for prosthetic nasal defects is three implants, which represent three points of attachment of the trabecular prosthetic structure. During qualification, it is crucial to perform a CT scan of the prosthetic area in order to be able to assess the quality and thickness of the bone necessary for the insertion of self-tapping implants. The titanium implants of the third generation Vistafix system (Cochlear, Goteborg, Sweden) have a diameter of 4.475 mm and are screwed in 3 or 4 mm deep. In our experience, however, the use of two Vistafix implants inserted into the alveolar bone of the maxilla also offers the possibility of stable fixation of the nasal prosthesis and its adhesion, although less tight compared to the three points of support.

Tso et al. described a case of external nasal reconstruction using Zygan Southern Implants titanium implants, which were mounted in two zygomatic bones, guided through the maxilla [12]. One of the implants was removed due to its instability four months after surgery, and a prosthesis was successfully fitted on only one implant. Southern Implants (South Africa) has been manufacturing dental implants for numerous years and has recently started producing titanium implants attached to the zygomatic bone, which can be used as a scaffold for facial prostheses [27]. It appears that they may be a good alternative for patients who underwent very extensive cancer surgeries and the condition of the facial bone does not allow implantation of the Vistafix system. Based on a recent study by Rogers et al., the procedure in the field of the reconstruction of craniofacial defects after extensive craniofacial surgeries has changed [9]. At present, more and more emphasis is placed on the primary installation of zygomatic implants, which are the skeleton for the addition of soft tissues of the craniofacial face with skin flaps. Zygomatic implants are a relatively new solution; however, there is a publication evaluating possible complications after their implantation. A lower rate of severe complications compared to other treatment options in extreme upper jaw atrophy was found in the 5-year follow-up [28].

Some reconstructive techniques of craniofacial deformities caused by non-Hodgkin’s lymphoma have been described. Osseointegrated implants are not used due to the aggressive nature of the tumor. Mainly magnets and tich buttons are used to attach the nasal prostheses [22].

The process of osseointegration of the cells on the surface of the titanium implant is central to the success of the implant and prosthetic surgery.

### 3.3. Contraindications

Absolute contraindications for implantation are a lack of oncological radicality and a patient’s condition that prevents proper implant hygiene due to severe mental disorders (e.g., dementia), cachexia, drug or alcohol addiction or a disability that makes it impossible to put on, take off and take care of the hygiene of the prosthesis. Relative contraindications are mild psychiatric disorders and poor craniofacial bone condition.

### 3.4. Implantation Procedure

The basics of the implant procedure were already defined by Brånemark (first implant procedure in 1979). The implantation process is carried out in several stages.

During the planning and qualification of the patient for such an operation, attention must be paid to the condition of the bone in which the implants are planned to be fitted. It is also very important to cooperate with a prosthetist, so that together they can plan the best location for the implants, both in terms of the stability in the bone and the fitting of the prosthesis. This is why several specialists are involved in the surgical planning stage: surgeons, clinical engineers, prosthetists and radiologists [29]. Pre-operative planning should include craniofacial MRI and CT scan analysis, 3D reconstructions of the craniofacial bones, clinical examination of the nasal deformity, endoscopy and rhinoscopy. Pre-establishment of implant sites is crucial, during which consideration is given to:-bone parameters—thickness and shape,-implant stability,-minimizing the impact of unwanted forces on the implants,-planning locations for replacement implants in case of implant failure at the originally selected sites, and-maintaining the angles at which the implants are inserted so that other prosthetic components can subsequently be fixed to them.

Nowadays, computer-aided design and manufacturing (CAD/CAM) are increasingly being used for this purpose. They allow implants to be virtually positioned in potential locations or models corresponding to the implanted fragments to be created and then printed using 3D technology for prior training [30,31].

During the patient’s qualification visit, the etiology of the defect (congenital, acquired after an accident or oncological) is taken into account. The site and method of prosthetic placement are determined, assuming a single- or two-stage operation, as well as the initial number of points required for stable mounting of the prosthesis.

The surgical treatment time depends on the number of surgical stages and loading time of the implant with an additional structural element such as the abutment, as well as the patient’s time of recovery. The completion of the surgical part is approx. two months after the last operation. This is the time when the skin around the implant should have completely healed. This will guarantee that the created prosthesis will fit well with the patient’s body.

Single-stage surgery can be opted for under a few conditions: when the bone is of good quality, the implant has been screwed correctly, the stability of the implant has been confirmed via Resonance Frequency Analysis (RFA) and a correct osseointegration process is expected. Single-stage surgery is most commonly used in prosthetic procedures for auricular defects.

Two-stage surgery is recommended if we expect a slower osseointegration process, in cases such as the following:-longer time required for soft tissue healing;-simultaneous removal of oncological lesions and insertion of anchored implants;-planned radiotherapy treatment.

If the operation is performed in a two-stage process, after insertion of the implant, a safety screw is tightened instead of the abutment, which emerges through the skin in a single-stage operation, and it serves to prevent contamination of the thread. Subsequently, the secured implant is sutured under the skin until the abutment is loaded. After a few months—minimum 3 months [11,32,33]—the implants are uncovered, the safety screws are removed and the abutments are inserted in their place, with the skin being abraded, as in single-stage surgery.

### 3.5. Silicone Prosthesis vs. 3D-Printed Prosthesis

The prosthetic part consists of the manufacturing and fitting of the prosthesis, individually to the patient, and starts around 7–8 weeks after implantation, once the soft tissue swelling has subsided. The prosthesis is moldable and adaptable to the correct shape and allows color matching to resemble the patient’s skin. It is made of silicone based on an acrylic plate, and the manufacturing process consists of four stages:-taking an impression of the area for the prosthesis,-fitting of the external structure developed from archival photographs of the patient’s face,-modelling the prosthesis in wax to obtain its final shape, and-acceptance of the prosthesis, learning to use the system.

Modern technical methods make it increasingly possible to replace hand-made prostheses with ones manufactured using 3D printing. However, a prosthesis created in this way may have less precision in adhering to the patient’s face. The problem mainly relates to marginal adaptation, which is caused by a too-thick material layer of 0.4 mm, meaning a serious limitation of the prosthesis printing technique compared to a thickness of less than 0.1 mm in conventional prostheses [18]. Improvements in manufacturing technology must address this point to achieve smoother transitions of the prosthesis edges into the patient’s skin. Therefore, at the current stage of development of silicone printing technology, the prosthesis is only suitable as a temporary post-operative solution. The latest publication achieved a 0.2 mm layer thickness for nasal mould printing, which is close to optimal value [34]. The second limitation of printed prostheses is the requirement for material quality. As intended, the prosthesis must be made of medical-grade (certified) silicone, which is not yet available for 3D printing [18].

### 3.6. Prognostic Factors Influencing Implantation Success

#### 3.6.1. Tissue Irradiation Dose

An important parameter that is relevant to the effects of implantation is the radiation dose received by the craniofacial region during radiotherapy [20]. Most researchers agree that a higher radiation dose increases the risk of implant prolapse. After radiotherapy, there is a marked reduction in bone vascularization and an imbalance between bone formation and resorption. It has been taken under consideration that blood vessels in bone regulate osteogenesis and haematopoiesis. Previous studies suggested that bone lacks lymphatic vessels, but a recent study has revealed the presence of lymphatic vessels in mouse and human bones [35]. Moreover, lymphatic vessels in bones support hematopoietic and bone regeneration. Thus, some authors mentioned that the radiation dose can impact bone regeneration, leading to the lower osseointegration capacity of implants. However, in a meta-analysis, this observation was not confirmed [17,36]. The authors studied the long-term effects of anchoring titanium implants in patients with nasal, ear and orbital prostheses according to radiotherapy exposure and found no statistically significant difference between groups. Some researchers suggest that, in most meta-analyses, the follow-up time for patients after implantation is too short (usually less than 5 years), hence the lack of clear differences in implant retention time between the group undergoing combined treatment with radiotherapy and the group undergoing surgical treatment alone [19]. According to some authors, higher implant failure rates are expected after 5 years [37]. Thus, it was concluded that, although not statistically significant, the durability of craniofacial implants is worse in patients after radiotherapy, especially for orbital region implants. However, better designed studies with detailed radiation doses and a longer follow-up period are needed. Therefore, in order to reduce problems with implant osseointegration after radiotherapy, it is worth to consider the primary placement of implants at the time of rhinectomy [12].

#### 3.6.2. Number of Implants and Their Interposition, Implantation Technique

Two to three anchorage points are required for the creation and good embedding of the prosthesis. The load and strength of each implant depend on their interposition, as well as the thickness of the bone into which they are inserted. Substrate factors are extremely important for proper osseointegration: bone quality, its density, vascularization, anatomical shape [38] as well as the surgical procedure itself, the angle of implant insertion or not overheating the bone when making the implant hole [37].

#### 3.6.3. Implant Site

The craniofacial region, particularly the midface, presents a surgical challenge for implant insertion.

As indicated in the literature, implants placed in the temporal bone area have a better prognosis of survival and fewer complications compared to other areas of the skull [39]. The highest risk exists in the nasal area [40]. According to some authors, the implant success rate in the nasal area also varies depending on the anatomical site in which the implant is placed (the implant success rate: 0% in the glabella and 88.1% in the anterior nasal floor, averaged at 71.4%). All implant failures occurred within the first year of loading [38].

#### 3.6.4. Patient’s Age

It is known that aging negatively affects organ function. The age at which osseointegration can be performed may vary depending on the specific procedure and the individual’s overall health. While osseointegration is possible at various ages, there are a few factors to consider such as bone density and quality, healing capacity and also systemic conditions.

Age-associated natural decline in bone density and quality can affect the ability of the bone to fuse with the implant. However, factors such as overall bone health and the specific condition of the patient’s bones will play a significant role. Tissue aging was investigated by some authors [36]. In a few organs, there was observed a loss of vessel density, and pericytes emerged as the mark of aging organs and tissues. However, highly remodelling tissues such as skin preserve the vasculature. Similarly, vessel densities remain unaffected in aging bones, which have relatively higher regeneration potential compared to tissues such as the kidney, spleen, heart or brain [41]. Nevertheless, age-associated vascular changes are only one aspect of osseointegration. Another factor is the ability of the body to heal and regenerate bone tissue. Generally, younger individuals tend to have better healing capacity compared to older individuals, which is an important aspect of successful osseointegration. Moreover, older patients may have a higher probability of having systemic conditions such as diabetes, osteoporosis or compromised immune systems, which influence the process of osseointegration and increase the risk of complications. Therefore, all those conditions have to be evaluated before a decision on implantation.

#### 3.6.5. Aftercare

Subcutaneous tissue reduction and split skin grafts are important in surgical aftercare. Over the years, clip repairs, fabrication of new prostheses, repair of the superstructure and fabrication of a new superstructure are needed [42]. The main problem after implantation is soft tissue infections around the implants. Therefore, daily hygiene of the implants is very important. As elderly patients may have difficulties in maintaining satisfactory hygiene, their family members or caregivers should be included in the prophylactic hygiene support program.

### 3.7. Own Experiences

#### 3.7.1. Case Report

In December 2021, in the Clinic of Otolaryngology CMKP in Warsaw, the first Polish single-stage procedure involving the placement of Vistafix 3 system titanium implants was successfully performed. The procedure led to effective fixation of a nasal prosthesis in a patient after the total amputation of the external nose and the anterior wall of the frontal sinus caused by squamous cell carcinoma of the maxilloethmoidal massif.

Ten months earlier, in February 2021, a 51-year-old patient was admitted to the clinic due to recurrent nosebleeds and a change in the shape of his external nose. Physical examination revealed a collapsed saddle nose and perforation of the nasal septum. A craniofacial CT scan confirmed a tumour of the nasal septum, bilaterally infiltrating the anterior ethmoid cells and frontal sinuses (Figure 2a). Histopathological examination of nasal samples confirmed squamous cell carcinoma. In March 2021, removal of the external nose with partial removal of ethmoid cells and the anterior walls of the frontal sinuses was performed, with subsequent radiotherapy completed in June 2021 (the patient received a radiation dose of 67.5 Gy) (Figure 2b). On completion of the oncological treatment, due to the inability of reconstruction with the patient’s own tissues as well as unsatisfactory results from the use of a spectacle-mounted prosthesis, the patient was qualified for the placement of osseointergrated implants for external nose prosthesis. Analysis of craniofacial CT scans in a multidisciplinary team identified three potential implant sites in the craniofacial bones. In December 2021, under general anaesthesia, the Vistafix 3 system implantation surgery was performed.

#### 3.7.2. Surgical Technique

The surgery was performed as a single-stage one. The thickness of the skin in the areas designated for the implants was measured in order to select the optimal length of the abutment (range between 3.5 mm to 7.5 mm) (Table 2). The skin was incised beyond the line of the implant, at a distance of approx. 1 cm. The periosteum was dissected at the site provided for the implant. Using a guide drill, a hole was drilled to the width of the implant—an Ostelle drill (Cochlear, Goteborg, Sweden) at 2000 rpm was used. Drilling was performed under water irrigation to prevent destruction of the osteocytes. Reducing the rotation to angular speed (torque) of 30 Ncm, depending on the quality and hardness of the bone, the implant was inserted into the prepared hole. The surgeon anchored three craniofacial implants (3 and 4 mm, respectively) (VX1300, Cochlear, Goteborg, Sweden).

Subsequently, soft tissue abrading was performed simultaneously to minimize the risk of skin flap displacement around the abutment and was followed by a skin thickness measurement (Table 2). The optimal abutment (VXA300, Cochlear, Goteborg, Sweden) was selected and tightened to the implant. A 5 mm diameter hole was made in the skin using a biopsy punch and the abutment was emerged. Skin sutures and a compression dressing were applied.

Two implants were fitted just below the bottom edge of the piriform aperture in the right and left upper jaws. The surgery was preceded by the process of computer planning the placement of implants and determining their location in relation to soft tissues and taking into account the aesthetic effects we want to achieve (Figure 3a–c). Due to the condition after removal of the frontal sinus wall, it was not possible to place a third implant in the glabella area, which would have given more stability to the prosthesis attachment. A third implant was placed halfway up the piriform aperture in the left maxilla (Figure 3d). Unfortunately, due to insufficient bone thickness in this area, despite the implant being left under the skin, it was unstable and was removed six weeks after implantation. Despite these technical and anatomical difficulties, it was possible to install the prosthesis in a stable manner on two implants.

The key element was to orient the implants at the proper angle: perpendicular to the bone, while at the same time optimal for further affixation of the prosthetic. This allowed maximum utilization of the bone surface and better mechanical stability, as well as facilitating the creation of a supporting structure for the actual prosthesis in subsequent prosthetic stages.

One week after the operation, the sutures were removed, and six weeks later, the implants were loaded with abutments and the patient was referred to the prosthetic laboratory. The total time from implant surgery to the placement of the final prosthesis was 14 weeks. The patient received a silicone prosthesis with an acrylic bearing part, in which the clips that fix it in place were embedded, as well as a trabecular scaffold, individually developed according to the patient’s anatomical features and the location of the anchored implants. The prosthesis is ultimately successfully supported by two implants embedded in the maxilla at the base of the nose.

#### 3.7.3. Follow-Up

At a follow-up visit 12 months after the operation, both the patient and the surgeon spoke very highly of the cosmetic effect of the prosthesis, which allowed the patient to return to full social function (Figure 4). The prosthesis adheres very well and is fully accepted by the patient. The condition of the soft tissues around the abutments was assessed at 0 according to the Holgers scale, which indicates very good skin condition, absence of soft tissue overgrowth and granulation [43].

## 4. Discussion

Surgical treatment of tumours of the maxilloethmoidal massif consists of two stages—a stage of resection of the pathological lesion and a stage of reconstruction to fill the defect and restore the best possible function of the nose and sinuses. Reconstructive possibilities depend on a lot of factors, the most important being the location and extent of the defect. Autological reconstruction with cartilage, muscle, subcutaneous tissue and skin grafts are attempted to fill the defect of the external nose. Due to the complexity of the external nose, this reconstruction is always challenging and sometimes becomes impossible to perform. For small facial defects up to 1 cm^2^, this is the optimal method with good results [44]. The use of autological grafts can be combined with the use of Medpor porous polyethylene, commonly used for facial skeletal reconstruction due to its biocompatibility and durability; however, based on systematic review, Medpor has a higher rate of extrusion and infection than autologous materials [21,45]. Unfortunately, in the case of patients who have had a significant part of their external nose removed or who have had it completely amputated, the results of this method are not satisfactory due to poor functional and cosmetic outcomes [16,44,46]. Both patients and specialists performing the reconstructions assessed the aesthetic effect of the prostheses much better than autologous grafts of the nose [16]. Additionally, the oncological follow-up of the patient requires good access to surgical sites, which often becomes difficult or even impossible after transplantation.

The Titanium Plate System “Epiplating” for Osseo-integrated Bone Anchorage of Craniofacial Prosthesis is an alternative system [7]. It consists of plates which are adapted individually to the bony contours and screws. This system does not need osseointegration to achieve maximal stability. The Epiplating System gives the possibility to attach the plates to bone distal to the irradiated site and more favourable mechanics of load spreading between multiple screws [15]. A similar system was used in the case of a 67-year-old man who underwent a left maxillectomy, excision of cheek lesion, rhinectomy, neck dissection and free flap closure. The custom-made prosthesis allowed for a magnetic nasal prosthesis alongside further dental rehabilitation, including the insertion of maxillary teeth implants into associated abutments [10]. The Epiplating System is more invasive than the Vistafix System, and the implantation is longer and needs to be performed under general anaesthesia. Moreover, the oncological follow-up is better in case of screw implants.

There are also titanium basal implants (“disk” or “lateral” or “basal” implants); they are also used in craniofacial reconstructions. They consist of a cylindrical part and a larger, cortically anchored base plate. Unlike traditional root-form implants (i.e., screw and blade implants), which are inserted vertically and primarily designed to be supported by trabecular bone, these implants are inserted from the lateral aspect of the host bone, providing multicortical support. Implants possess one to three very pronounced “threads“ or “base-plates”, which are securely anchored in the cortical bone, a bone area which is more stable during the remodelling/resorption process and which can respond successfully to immediate loading protocols. The site of force transmission is far away from the site of bacterial invasion, allowing for early loading and resistance to infection. In Konstantinović et al.’s study, the survival rate of basal implants used for the retention of extraoral prostheses was assessed [8]. The overall 12-year survival rate of nasal basal implants (28 patients were included) was 92.9%, which is a very good result compared with screw implants. Further studies are needed on this type of implant to assess its usefulness in craniofacial reconstructions.

A non-invasive, widely used method of covering facial defects is silicone nasal prostheses mounted on spectacles or attached to the skin with silicone-based adhesives. The prosthesis is removed along with the glasses, giving easy access to inspect the post-operative wound, but its unstable fixation makes it significantly more difficult for a patient to return to social life. Loose attachment of the prosthesis to the face causes exhaled air to escape around the prosthesis, exposing the facial defects. The high risk of the prosthesis shifting and slipping when mounted on glasses causes patients to have limitations in physical activities, sports and some occupations, making it often difficult to accept [47]. It is also difficult to attach the prosthesis in the correct position with skin adhesives. Adhesives can cause skin irritation, allergic reactions and deterioration of the edges of the silicone prostheses [42].

Recent technical advances have allowed the introduction of further solutions in the reconstruction of defects in the midface, which are titanium implants anchored in the bone, allowing effective and stable fixation of the silicone prosthesis. These implants are subject to further improvements. According to studies, third-generation Vistafix facial implants have higher stability compared to earlier generations [48] and have a higher survival rate [49]. The new materials reduce soft tissue reaction around implants placed in the anterior part of the nose [38]. Changes in surgical technique, simplifying it, have significantly reduced the rate of adverse skin reaction [14,29,37,48].

The modification of the operation to a single-stage procedure that we performed in our clinic twice shortened the time needed for final fixation of the prosthesis, which was 14 weeks instead of 6–8 months for a two-stage operation [14,50,51].

The case we presented also confirms the possibility of stable fixation of the prosthesis on two 4 mm long implants placed at the anterior nasal floor, which is the most commonly used location for this type of reconstruction [38]. Two implants allow for a very good cosmetic result and provide a chance for reconstruction in patients in whom it is not possible to attach a third implant for technical reasons, e.g., proximity to the bone suture, insufficient bone thickness, proximity to the maxillary sinus, inability to completely screw the implant into the bone or inability to obtain an optimal implant insertion angle.

In addition to the aesthetic benefits, significant advantages of this method of reconstruction are the improvement in the articulation of the patient’s speech, resulting in better verbal contact; protection of the upper respiratory tract against aspiration of foreign bodies and easy oncological control.

We would like to emphasise that implants do not conflict with plastic reconstructive methods and can often be used complementarily or as a temporary tool prior to surgical reconstruction (temporary provision). This is especially important for patients who have had their external nose removed, where a comprehensive reconstruction can only be carried out after two years, when the risk of recurrence is significantly reduced [47]. Implantation does not impede subsequent reconstruction. This also makes implanted prostheses more than just an alternative.

The results of the studies included in this review are limited by the relatively small number of patients sampled for the studies, allowing only qualitative analysis. Within the limitations of this study are the small groups of patients with various types of implants, the limited number of adequate studies and the fact that patients who were qualified for nose reconstruction had undergone oncological operations of various scope, which significantly affected later possibilities of choosing the implantation site and, thus, the prognosis as to the maintenance of implants. This is similar to our patient, for whom the scope of the operation was so extensive that we were not able to place the third implant properly.

## 5. Conclusions

Anchored craniofacial implants are a good and minimally invasive solution for postnasal resection patients with a low rate of adverse reactions. In selected clinical cases, stable fixation of the prosthesis on two anchored implants is possible and their single-step implantation significantly speeds up the prosthetic process.

## Figures and Tables

**Figure 1 healthcare-11-01663-f001:**
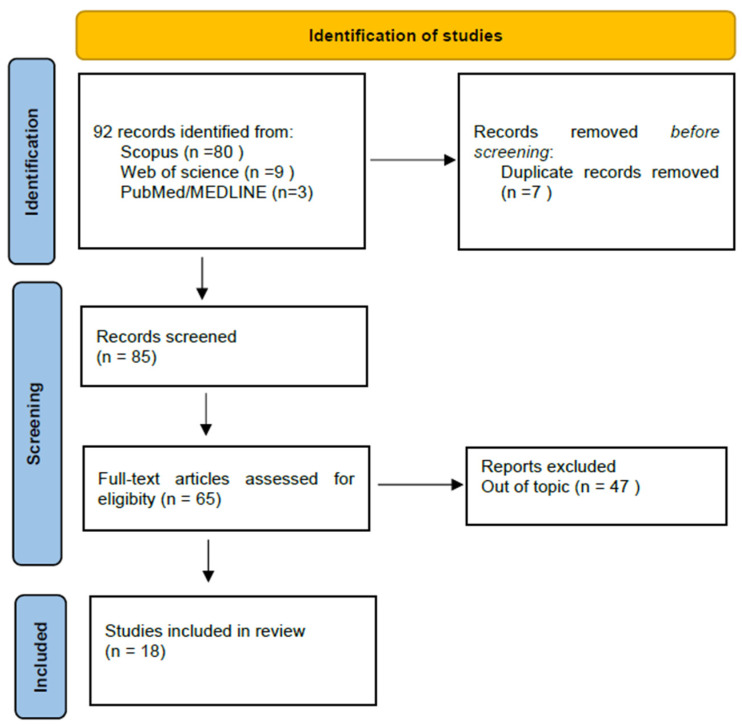
Flow diagram of the systematic literature search.

**Figure 2 healthcare-11-01663-f002:**
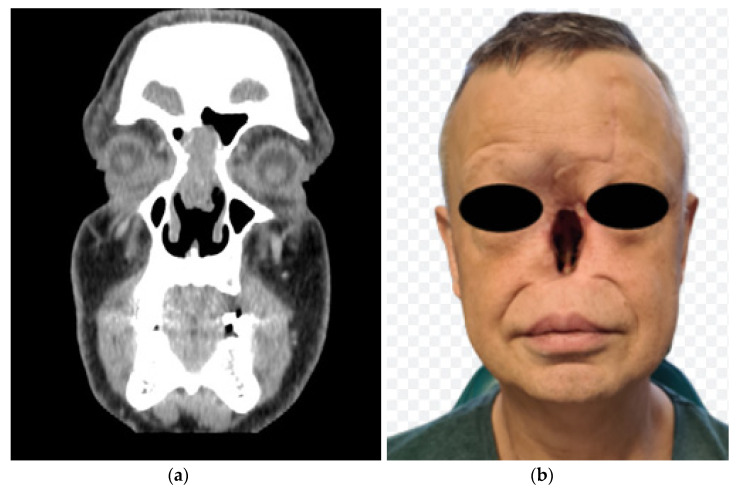
(**a**) Craniofacial CT scan—the tumour infiltrates nasal septum, ethmoidal cells and frontal sinuses. (**b**) Patient after removal of the tumour.

**Figure 3 healthcare-11-01663-f003:**
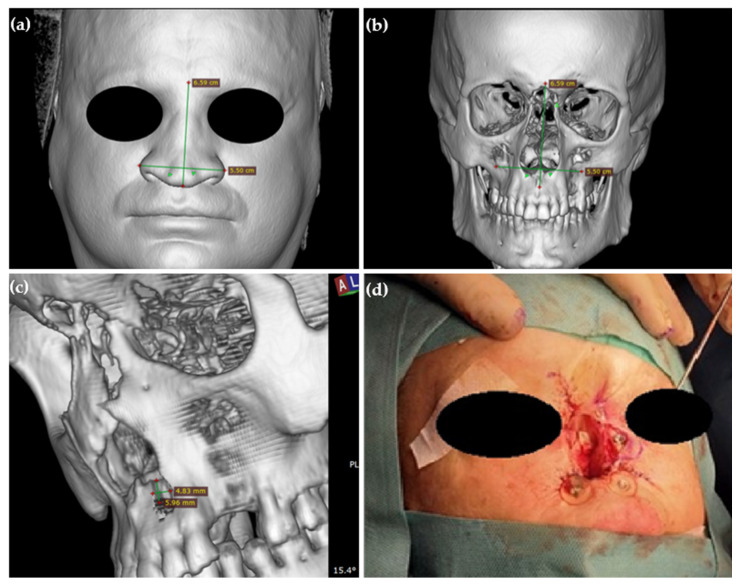
(**a**) Nose dimensioning; CT image, 3D imaging skin mode. (**b**) Planning places for implants takes into account the dimensions of the nose. (**c**) Determining the thickness and width of the bone and the angle of inclination at the planned implant placement site. (**d**) Photo of mounted implants intraoperatively.

**Figure 4 healthcare-11-01663-f004:**
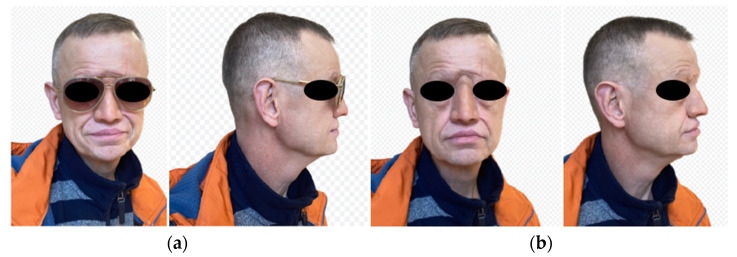
Patient after epithesis (**a**) with glasses and (**b**) without.

**Table 1 healthcare-11-01663-t001:** Characteristics of studies included in the review.

		Number of Patients with Craniofacial Implants	Number of Patients with Nasal Prostheses	Type of the Implant
1.	Wrobel C. et al. (2022) [7]	1	1	Medicon Epiplating system
2.	Konstantinović S. et al. (2022) [8]	26	11	Disc implants
3.	Rogers S. et al. (2022) [9]	12	6	Zygomatic implants
4.	Gunawardena D. et al. (2022) [10]	1	1	Implant abutments KLS Martin self-screwing screws
5.	Alberga J. et al. (2022) [11]	201	45	The Nobel Biocare and Entific Medical Systems
6.	Tso T. et al. (2021) [12]	1	1	Zygomatic implants
7.	Gaur V. et al. (2021) [13]	1	1	Zygomatic implants
8.	Rosen E.B. et al. (2020) [14]	1	1	Vistafix System
9.	Nowak S. M. et al. (2020) [15]	1	1	Medicon Epiplating system
10.	Dings J. P. J et al. (2020) [16]	41	24	No data
11.	Moore P. et al. (2019) [17]	54	13	Vistafix System, Institut Straumann AG implants
12.	Unkovskiy A. et al. (2018) [18]	1	1	Vistafix System
13.	Subramaniam S. S. et al. (2018) [19]	110	15	Vistafix System, Institut Straumann AG implants, The Nobel Biocare and Entific Medical Systems
14.	Papaspyrou G. et al. (2018) [20]	99	19	Medicon implant System, Vistafix System, Leibinger System, Straumann System
16.	Khorasani M. et al. (2018) [21]	16	11	Medpor porous polyethylene implants
17.	Dings J. et al. (2018)	38	14	No data
18.	Bansal K. et al. (2018) [22]	2	2	Magfit DX

**Table 2 healthcare-11-01663-t002:** Size of the used third-generation Vistafix implants and abutments depending on the implant site.

No.	Implant Site	Skin Thickness before Abrading	Skin Thickness after Abrading	Implant VX1300 Length	Abutment VXA300 Length
1.	Maxillary bone, nose floor, right side	7 mm	5 mm	4 mm	7.5 mm
2.	Maxillary bone, nose floor, left side	6.5 mm	4.5 mm	4 mm	7.5 mm
3.	Frontal process of maxilla, left side	3 mm	2 mm	3 mm	4.5 mm

## Data Availability

Not applicable.

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
