# Peer review of "The Latest Craniofacial Reconstructive Techniques Using Anchored Implants after Surgical Treatment of Nasal and Paranasal Sinuses Tumors"

_healthcare, 2023, doi:10.3390/healthcare11121663_

Round 1

Reviewer 1 Report

The reference list of the manuscript contains 34 titles, and is without inappropriate self-citations. Three references are elder than 20 years. The manuscript is clear, with a moderate rate of novelty but a high clinical significance. The manuscript is not a review of current literature. It's more like a case report. It is well investigated, but the discussion section should be supplemented to aspects such as epithesis plates.  (E.g.https://medicon.de/produkte/transversale-ti-nasenplatte/   ). The main advantage of implants over epithesis plates is oncological aftercare and easy maintenance.

LL169 – 171

After a few months … Please specify. Out of my experience, 6 weeks are sufficient for covered healing at the orbit, ear and nose.

Addition: This manuscript is not a review. It is just a case report of a single institution. It does not give an overview of the possible reconstruction procedures. Thus, epithesis plates should be mentioned in any case. In particular, the paper should be completed with a look at patient age, surgical-technical and oncological aspects. Age is a central aspect and often tips the scales when it comes to autologous reconstruction, for example, with paramedian forehead flap in combination with cartilage grafts. Since the manuscript is not a review paper, there is no material and methods part. In this case report, there should at least be a clearer and more structured discussion of which inclusion and exclusion criteria the authors choose for one procedure or another. I miss that in this paper and it is only very superficially dealt with in section 2.3. In addition, I would recommend a critical discussion of the ideal implant position. Also missing are aspects of epithesis and implant care. After all, these contribute to success in the long term.

Author Response

Reviewer’s comment 1:

The discussion section should be supplemented to aspects such as epithesis plates.  (E.g.https://medicon.de/produkte/transversale-ti-nasenplatte/). The main advantage of implants over epithesis plates is oncological aftercare and easy maintenance.

Answer:

Thank you for sharing this observation and for recommending Epiplating System to supplement the discussion. As suggested, in the discussion section, we have added information about the Epiplating System and other new solutions- lines 418-444.

Reviewer’s comment 2:

This manuscript is not a review. It is just a case report of a single institution. It does not give an overview of the possible reconstruction procedures. 

Answer:

Thank you for this comment. We did the systematic literature search following PRISMA. We have added this information in materials and methods section with flow diagram of the systematic literature search, list of publications included in our review and we expanded our overview of the possible reconstruction procedures (text added in Discussion).

The article has been enriched with 18 new publications from the last 5 years. We hope that these changes meet your expectations.

Reviewer’s comment 3:

In particular, the paper should be completed with a look at patient age, surgical-technical and oncological aspects.

Answer:

We have added paragraph about implant aftercare- lines 305-312 and patient's age- lines 282-303.

Reviewer’s comment 4:

There is no material and methods part.

Answer:

We have added materials and methods section. We apologize for lack of this section in the first version of the manuscript. We are sure this suggestion will improve the manuscript.

Reviewer’s comment 5:

In this case report, there should at least be a clearer and more structured discussion of which inclusion and exclusion criteria the authors choose for one procedure or another.

Answer:

We improved the discussion organization as suggested and added the manuscript limitation. We also added inclusion and exclusion criteria in Material and Methods section.

Reviewer’s comment 6:

 I miss that in this paper and it is only very superficially dealt with in section 2.3. Also missing are aspects of epithesis and implant care. 

Answer:

We added information about epithesis and implant care - lines 305-312

Reviewer’s comment 7:

After a few months … Please specify. Out of my experience, 6 weeks are sufficient for covered healing at the orbit, ear and nose.

Answer:

Thank you for this comment. We added the information in line 238 - Most of authors suggest that second-stage operation should be performed minimum 3 months after implantation.

We thank you for your thorough and expert review and hope that the changes strengthened the manuscript.

Reviewer 2 Report

This is a very interesting manuscript on reconstructive surgery after surgical treatment of neoplasms in the head and neck region. The complexity of the facial anatomy affects the aesthetic outcome of the reconstruction, and postoperative radiotherapy limits the range of reconstructive techniques. The study focuses on current reconstructive methods in the craniofacial region, specifically the use of bone-anchored implants to attach nasal prostheses. A successful single-stage, Vistafix 3 osseointegrated implant for connecting an external nasal prosthesis in a 51-year-old man after surgical removal of squamous cell carcinoma of the nose and paranasal sinuses. The results are robust and significant.
Vasculature plays a critical role in tissue and bone regeneration PMID: 36669473; PMID: 33536212. Authors should include and cite these and other relevant work. Also, the authors should discuss the impact of age on the outcome.

Author Response

Reviewer’s comment :

Vasculature plays a critical role in tissue and bone regeneration PMID: 36669473; PMID: 33536212. Authors should include and cite these and other relevant work. Also, the authors should discuss the impact of age on the outcome.

Answer:

Thank you very much for this comment. Vasculature and age are indeed very important aspects in the context of craniofacial implants. We added new paragraphs about the age and vasculature- lines 308-329 and 271-276. We added these references to our article. We hope that the changes strengthened the manuscript.

Reviewer 3 Report

Interesting study about the implants for nasal and paranasal tumors reconstruction. There are some interesting references and the surgical description is enough. The authors describe the literature and their own experience, however, I would like to make some comments:

- the material and methods section is missing. The authors should include it and especify how they did the review of the literature and how they presented their results.

- It would be necessary to also include limitations at the end of the discusion.

I found this paper very interesting, especially the author’s experience. But I did not find the manuscript scientific enough if the authors do not explain how they made the review of the literature, and they did not include limitations about their article.

Author Response

Reviewer’s comment 1:

The material and methods section is missing. The authors should include it and especify how they did the review of the literature and how they presented their results.

Answer:

Thank you very much for this review. We have added materials and methods section - lines 54-74 and we have specified the search strategy and eligibility criteria.

We did the systematic literature search following PRISMA. We have added this information in materials and methods section with flow diagram of the systematic literature search, list of publications included in our review. We apologize for lack of this section in the first version of the manuscript.  

Reviewer’s comment 2:

It would be necessary to also include limitations at the end of the discussion

Answer: We have added the limitations at the end of the discussion lines 510-517.

Round 2

Reviewer 2 Report

Authors have addressed all my comments. I have no further comments.

Reviewer 3 Report

congratulations to the authors. The new information added to the manuscript, and especially the description of the review of the literature as a PRISMA has increased the quality other paper. In my humble opinion this study is now suitable for the publication in the journal.